# Revealing the Immune Response of *Sitona callosus* Gyllenhal to Entomopathogenic Fungi *Beauveria bassiana* Infection Through Integrative Analyses of Transcriptomics and Metabolomics

**DOI:** 10.3390/insects15120940

**Published:** 2024-11-28

**Authors:** Nan Li, Xin Gu, Ming Xin, Xinpu Wang

**Affiliations:** 1School of Forestry and Prataculture, Ningxia University, Yinchuan 750021, China; linan950029@163.com; 2School of Agriculture, Ningxia University, Yinchuan 750021, China; isaacxin@163.com

**Keywords:** *Beauveria bassiana*, *Sitona callosus*, RNA-Seq, metabolomics, immune response

## Abstract

The legume pest *Sitona callosus* represents a significant threat to alfalfa cultivation. Due to its unique life cycle, *Beauveria bassiana* has emerged as an effective entomopathogenic fungus for controlling *S. callosus*. We performed a comprehensive analysis of the transcriptome and metabolome of *S. callosus* infected with *B. bassiana.* The differential expression of several antifungal genes, including heat shock proteins, cytochrome P450 (CYP450) enzymes, cathepsin proteases, and C-type lectins, was observed in *S. callosus*. A pathway analysis revealed significant associations between immune-related genes and metabolites involved in autophagy, the glucagon signaling pathway, and glycerophospholipid metabolism. These findings provide valuable insights that could enhance the efficacy of *B. bassiana* in controlling *S. callosus*.

## 1. Introduction

*Sitona callosus* (Coleoptera: Curculionidae: Bruchinae) is a significant pest on leguminous plants in numerous countries [1,2,3,4]. Both adult and larval root weevils inflict damage upon leguminous plants. During the early stages of plant emergence, adults feed on tender shoots, resulting in the degeneration of the growing point or the death of young shoots. In mature plants, adults consume leaves and severe infestations can lead to complete defoliation [5]. The larvae exclusively feed on the root nodules, leading to a reduction in their numbers. Consequently, nitrogen absorption is impeded and the wounds caused by larval biting become more susceptible to microbial invasion, resulting in root rot and subsequently hampering plant growth and development [6,7,8,9,10]. Research has demonstrated that during years with high weevil densities, root weevil larvae can destroy over 90% of nodules, leading to stunted growth or plant mortality due to nutrient deficiency [11]. Due to the unique life history of *S. callosus*, the larvae inhabit the rhizosphere soil of the plant, posing challenges for effective control measures. Therefore, it is crucial to consider soil pesticide residues and forage safety while ensuring optimal control efficacy. Consequently, addressing biocontrol strategies for managing *S. callosus* has emerged as a significant concern in agricultural production.

In recent years, the excessive use of chemical pesticides has progressively bolstered pest resistance, posing challenges for conventional chemical agents to achieve the desired control efficacy. Consequently, biological control has emerged as a pivotal approach in pest management. In particular, the utilization of entomopathogenic fungi plays a crucial role in reducing pest population density within integrated pest management (IPM) programs [12]. The development and application of a plethora of entomopathogenic fungi have been undertaken to mitigate reliance on chemical pesticides in pest management [13]. Entomopathogenic fungi naturally occur and are non-toxic and harmless, posing no risk to human or livestock health [14]. Instances of pest resistance to entomopathogens are infrequent [15]. *Beauveria bassiana* is a broad-spectrum EPF that induces white muscardine disease in insects such as whiteflies, aphids, Tasseloptera, Orthoptera, and Coleoptera [16]. *B. bassiana* exhibits significant potential as an entomopathogenic fungus for pest management purposes.

Transcriptomic applications in insects have emerged as a pivotal area of investigation for comprehending the genetic mechanisms and responses of insects to diverse stimuli. Transcriptome analysis enables researchers to explore the intricate patterns of gene expression and regulatory networks that govern insect physiology and behavior, facilitating a more sophisticated understanding in the field. Some researchers utilized RNA sequencing data to generate a de novo transcriptome assembly of *Trichoplusia ni* in response to infection by the baculovirus *Autographa californica* multiple nucleopolyhedrovirus. The differential gene expression analysis revealed the significant downregulation of host transcripts post infection, providing insights into the host–virus interaction dynamics [17]. In a comprehensive transcriptome analysis, key genes involved in various biological systems were identified in an olive fly study, highlighting the crucial role of chemical insecticides in controlling olive fly populations [18]. Furthermore, an investigation on mevalonate farnesyl biosynthesis in ticks revealed the conservation of enzyme regions and motifs within the biosynthetic pathway [19]. Transcriptome analysis has emerged as a potent tool for elucidating the intricate facets of insect biology and comprehending their interactions with the environment, pathogens, and compounds; however, it still exhibits certain limitations. In recent years, the advancement of sequencing technology has led to an increased integration of transcriptomics with proteomics and metabolomics, thereby complementing each other. The utilization of transcriptome–metabolome association in insects has garnered significant attention owing to its potential for comprehending insect resistance mechanisms and devising strategies for pest management. The transcriptome and metabolome of *Ostrinia furnacalis*, a significant maize pest, were compared to investigate the immune response upon UV-A exposure. The findings demonstrated substantial alterations in signal transduction, antioxidant defense mechanisms, and the insect immune system following ultraviolet irradiation. Moreover, the metabolism of amino acids, sugars, and lipids was also impacted [20]. These results elucidate the mechanism underlying insect adaptation to environmental stressors and provide valuable insights for studying insect immunity. In another study focusing on *Drosophila melanogaster*, researchers examined the effects of cold acclimation on both transcriptomic and metabolomic profiles. The analysis revealed major recombination in gene expression patterns and metabolic pathways during cold acclimation. Notably, the proline and glutathione metabolic pathways exhibited strong correlations with enhanced cold tolerance [21]. Transcriptional and metabolic analyses showed that the immune recognition and antimicrobial peptide gene of Thitarodes xiaojinensis upregulation were the most obvious in response to *Ophiocordyceps sinensis* infection [22].

In order to investigate the *Sitona callosus* gene information associated with the pattern recognition, signal transduction, regulation, and defense response in *Beauveria bassiana* infection, we employed a combination of transcriptomics and metabolomics to examine the impact of *B. bassiana* on gene expression and metabolic cycles of *S. callosus*. We identified possible immune pathways and screened key differentially expressed genes involved in immune responses. These findings will facilitate future biological control strategies for *S. callosus* in *Medicago sativa* (alfalfa), leading to improved yield and quality.

## 2. Materials and Methods

### 2.1. Strains and Insects

The *Beauveria bassiana* strain B1 used in this study was isolated from the cadaver of *Sitona cylindricollis*, a pest of alfalfa, and showed high virulence against the pest in preliminary tests [23]. The laboratory tests also revealed a high virulence of the pathogen towards *Sitona callosus*. The strain is maintained in the plant protection laboratory of Ningxia University. Experiments were conducted using a laboratory-reared population of *S. callosus*.

### 2.2. Spore Suspension and Preparation of S. callosus Samples

*Beauveria bassiana* was cultured on PDA medium for approximately 10 days. Petri dishes, after being inoculated with *B. bassiana*, were placed in light incubators (temperature: 28 °C, humidity: 80%, light-dark cycle: 16 L:8 D). Spores were harvested using a sterile loop and suspended in 500 mL of sterile water containing 0.01% Tween-80 to achieve spore concentrations of 10^4^, 10^5^, 10^6^, 10^7^, and 10^8^ spores/mL. *S. callosus* samples were prepared using a “dipping insect + dipping leaf” method. Healthy and uniform-sized adult *S. callosus* were carefully immersed in the spore suspension using sterile forceps, ensuring full contact with the suspension. After 10 s, the excess spore suspension was removed using absorbent paper. Fresh alfalfa leaves, wrapped with damp sterile cotton at the petiole to maintain freshness, were then dipped into the spore suspension for approximately 10 s and allowed to air dry. Throughout the sampling process, *S. callosus* were fed with alfalfa leaves treated using this method. For the control group, the insects were dipped in sterile distilled water containing Tween-80 for 10 s. Each treatment group consisted of 60 individuals, with 3 replicates. Three groups were established: T48, sampled 48 h post infection; T96, sampled 96 h post infection; and T144, sampled 144 h post infection. After collection, the samples were immediately frozen in liquid nitrogen and stored at −80 °C for subsequent transcriptomic and metabolomic analyses.

The potato dextrose agar (PDA) utilized in the aforementioned study was procured from Haibo Bio, Qingdao, China (item number HB0233-12). Its primary constituents consist of potato extract powder (12.0 g/L), glucose (20.0 g/L), and agar (14.0 g/L), with a pH of 5.6 ± 0.2. Preparation method: Weigh 46.0 g of the product and dissolve it in 1000 mL of distilled water. Autoclave the solution at 121 °C for a duration of 30 min, followed by setting it aside.

### 2.3. Experimental Method and Analysis Process of RNA Sequencing from S. callosus

Total RNA was extracted from *S. callosus* using a total RNA isolation kit. The mirVana™ miRNA isolation kit (Cat#. AM1561, Ambion, Austin, TX, USA) from Thermo Fisher Scientific was employed as the total RNA isolation kit in this study. The Illumina Novaseq 6000 sequencing platform was used. The Illumina Novaseq 6000 sequencing platform offers an output range of 80 to 6000 Gb, with the capability of generating between 650 million to 2 billion single-end reads per run. It supports a maximum read length of 2 × 250 bp and has a run time of approximately 13 to 44 h. Raw data (raw reads) in fastq format were first processed using Trimmomatic [24]. After removing adaptor and low-quality sequences, the clean reads were assembled into expressed sequence tag clusters (contigs) and de novo-assembled into transcripts using Trinity [25] (version: 2.4) with the paired-end method. The longest transcript was chosen as a unigene based on the similarity (≥80%) and length for subsequent analyses. After annotation, the software bowtie 2 [26] was used to obtain the number of reads aligned to the unigene in each sample, and the express [27] software was used to calculate the expression level of the unigene (FPKM [28]). Differentially expressed genes (DEGs) between different groups were identified using DESeq2 [29] software version 1.40.2 to calculate the fold of difference, and NB (negative binomial distribution test) was used to test the significance of difference; finally, the default filter conditions for DEGs was q < 0.05 and fold change >2 or fold change <0.5. A hierarchical cluster analysis of the DEGs was performed using R (v 3.2.0) to demonstrate the expression pattern of the unigenes in different groups and samples. GO enrichment and KEGG pathway enrichment analysis of the DEGs were performed, respectively, using R (v. 3.2.0) clusterprofile package [30] based on the hypergeometric distribution. R (v. 3.2.0) ggplot2 package [31] was used to draw the column diagram and bubble diagram of the significant enrichment term.

### 2.4. Widely Untargeted Metabolomics Analysis of S. callosus

Widely untargeted metabolomics analysis was performed to investigate changes in *S. callosus*-associated immune metabolite accumulation under different *Beauveria bassiana* infection times. The metabolite analysis of *S. callosus* infected with *B*. *bassiana* was performed by Shanghai Luming Biotechnology Co., Ltd. (Shanghai, China).

The sample was thawed on ice, and 20 mg was weighed into a 1.5 mL EP tube. Two small steel balls were added, followed by 400 μL methanol–water (*v*/*v* = 4:1, including a mixed internal standard at 4 μg/mL). After pre-cooling at −40 °C for 2 min, the sample was ground in a grinder (60 Hz, 2 min). Ultrasonic extraction was performed in an ice water bath for 10 min, and the sample was left overnight at −40 °C. Following this, the sample was centrifuged for 10 min (12,000 rpm, 4 °C). A 150 μL aliquot of the supernatant was extracted with a syringe, filtered through a 0.22 μm organic phase filter, transferred to an LC sample vial, and stored at −80 °C until LC-MS analysis.

A quality control (QC) sample was prepared by mixing equal volumes of the extracts from all samples. LC-MS chromatography was performed using an LMS system consisting of a Waters ACQUITY UPLC I-Class Plus/Thermo QE ultra-high-performance liquid chromatography–tandem high-resolution mass spectrometer.

The ACQUITY UPLC HSS T3 column (100 mm × 2.1 mm, 1.8 µm; waters) was used for reverse-phase separation, with the column temperature maintained at 45 °C. The mobile phases were as follows: A-water (0.1% formic acid) and B-acetonitrile (0.1% formic acid). The injection volume for each sample was 5 μL, and the flow rate was set to 0.35 mL/min. The gradient elution conditions were as follows: 5% solvent B (0–2.0 min); 5%-100% solvent B (2.0–14.0 min); 100% solvent B (14.0–15.0 min); 100%-5% solvent B (15.0–15.1 min); 5% solvent B (15.1–16.0 min). The original data were then baseline-filtered, peaks identified, integrations performed, retention times corrected, peak alignment conducted, and normalization applied using the metabolomics processing software Progenesis QI v3.0 (Newcastle, UK). A multivariate statistical analysis was initially performed using unsupervised principal component analysis (PCA) to observe the overall distribution among the samples and assess the stability of the analysis process. Subsequently, supervised partial least squares discriminant analysis (PLS-DA) and orthogonal partial least squares discriminant analysis (OPLS-DA) were applied to identify differences in the metabolic profiles between groups and pinpoint differential metabolites. Multidimensional and unidimensional analyses were employed to screen for differential metabolites between groups. We assessed the systematic stability of the PCA results. In the PCA model diagram generated through 7-fold cross-validation (comprising 7 cycles of cross-validation), the QC samples were found to be closely clustered together. This observation indicates that the experiment demonstrates good stability and repeatability. In the OPLS-DA and PLS-DA analyses, the variable importance in projection (VIP) score was used to assess the influence and interpretative capacity of metabolite expression patterns in the classification and differentiation of samples. This approach helped identify differential metabolites with biological significance. A T-test was further applied to verify the statistical significance of the differences in the metabolites between groups. Metabolites with a fold change ≥ 2 or ≤ 0.5 and a VIP ≥ 1 were considered to be differential metabolites (DAM). In differential metabolomics, the VIP value (Variable Importance in Projection) is a metric used to evaluate the correlation between a metabolite and sample classification. A higher VIP value indicates a greater contribution of the metabolite to the classification of the samples. The VIP value is calculated using a partial least squares regression model, taking into account both the model’s prediction error and the importance of the variables. Typically, in differential metabolomics, metabolites with VIP values greater than 1 are considered significantly different, allowing for further analysis of their biological significance and potential mechanisms of action.

### 2.5. Real-Time Fluorescence Quantitative PCR Detection

After total RNA extraction, the RNA concentration and OD260/OD280 ratio were determined using a NanoDrop 2000 spectrophotometer (Thermo Fisher Scientific, Waltham, MA, USA). The integrity of the RNA was assessed using agarose gel electrophoresis. The RNA was then reverse-transcribed into complementary DNA (cDNA) using the TransScript All-in-one first-strand cDNA synthesis super MIX for qPCR kit.

Six differentially expressed genes, including C-type lectin 37Db-like (TRINITY_DN19747_c0_g4_i1_1), PGRP (TRINITY_DN29537_c0_g1_i2_1), perlucin-like (TRINITY_DN15347_c0_g1_i2_1), GNBP3 (TRINITY_DN31229_c0_g1_i12_3), SCARB1 (TRINITY_DN31299_c1_g3_i2_1), NOX5 (TRINITY_DN31950_c1_g1_i1_4) were detected using fluorescence quantitative PCR. The ATG7 (TRINITY_DN30481_c0_g1_i6_4), which exhibited stable expression throughout the infection, was selected as the internal reference gene.

The gene expression levels were calculated using the 2^−∆∆Ct^ method [32]. Details regarding the DEGs, internal reference genes, and primers used for validation can be found in the Appendix A.

### 2.6. Statistical Analysis

The experimental data were organized using Microsoft Excel 2016. The mortality rates obtained from the indoor bioassay were corrected using Abbott’s formula [33]. The mortality rates of *S. callosus* treated with varying concentrations of spore suspensions were plotted based on the average mortality rates and standard errors obtained from the treatments, utilizing Origin 2024b.

## 3. Results and Analysis

### 3.1. Changes in S. callosus Infected with B. bassiana

The mortality of adult *S. callosus* varied with different concentrations of the spore suspension (Figure 1). However, starting from the sixth day post infection, there was a sharp increase in mortality. To gain deeper insights into the genetic and metabolic changes occurring in infected adult *S. callosus*, we selected adult *S. callosus* treated with a lower concentration of spore suspension for 48 h, 96 h, and 144 h as experimental subjects.

### 3.2. Transcriptome Data Quality Control and Analysis

High-throughput sequencing was employed to conduct a transcriptome-level analysis of 12 samples from four groups of *S. callosus*. The transcriptome data for the 12 samples are presented in Appendix A. A total of 83.58 Gb of clean data were obtained, with effective data volumes ranging from 6.86 to 7.06 Gb per sample and Q30 base ranging from 96.76% to 97.21%. The average GC content of all samples was 39.31%, indicating high accuracy. The obtained clean reads were aligned to Unigene, with alignment rates ranging from 90.19% to 90.82%. Trinity [25] software was used for assembly, and the results are shown in Appendix A, In total, 57,302 unigenes were obtained, with an average N50 length of 1238. There were 36,324 unigenes with lengths greater than or equal to 500 bp and 16,722 unigenes with lengths greater than or equal to 1000 bp. The longest unigene was 24,615 bp, the shortest was 301 bp, and the average length of the unigenes was 942.61 bp. These results show the accuracy of the transcriptome data and support further analysis.

To assess the repeatability of the transcriptome data for *S. callosus*, PCA and cluster analysis were performed on the 12 samples (Figure 2). In Figure 2a, groups T48-1, T48-2, and T48-3 are clustered together; groups T96-1, T96-2, and T96-3 are clustered together; and groups T144-1, T144-2, and T144-3 are clustered together. T48-1, T48-2, and T48-3 represent the three groups of *S. callosus* infected with *B. bassiana* for 48 h, T96-1, T96-2, and T96-3 represent the three groups infected for 96 h, and T144-1, T144-2, and T144-3 represent the three groups infected for 144 h. In Figure 2b, the samples from the three groups are clustered. Both the principal component analysis and cluster analysis results indicate the good repeatability of the samples for further analysis.

### 3.3. Gene Function

The assembled unigenes were annotated in various databases (Appendix A). A total of 23,626 genes (41.23%) were annotated to the NR database, 14,908 genes (26.02%) to the Swiss-Prot database, 7144 genes (12.47%) to the KEGG database, 13,665 genes (23.85%) to the KOG database, 19,408 genes (33.87%) to the eggNOG database, 13,832 genes (24.14%) to the GO database, and 15,991 genes (27.91%) to the Pfam database.

### 3.4. Differential Gene Expression Analysis

The differential genes of four groups of *S. callosus* were analyzed. The read counts among the different groups of data were standardized among all 57,302 unigene samples. DEGseq was used to analyze differentially expressed genes among the four groups of samples. When the q-value < 0.05 and |log2FC| > 1.0, we considered that this gene was significantly different among the groups in order to select less differentially expressed genes. Compared with the control (ck), the differentially expressed genes of the T48, T96, and T144 groups contained 4188, 3387, and 9045 unigenes, respectively.

According to the comparison of the differential gene expression between the treatment group and the control group, as shown in Figure 3 (blue represents downregulation, red represents upregulation), 2674 genes were upregulated and 1514 genes were downregulated at 48 h of infection (T48) compared with ck. At 96 h (T96), 2046 genes were upregulated and 1341 genes were downregulated compared with ck. Compared with ck, 5213 genes were upregulated and 3832 genes were downregulated at 144 h (T144). It can be concluded that the upregulated expression of genes in *S. callosus* became more active with the change in infection time. This may be related to the activation of the immune response and the enhancement of the overall metabolic level.

### 3.5. Enrichment Analysis of Differentially Expressed Genes: GO and KEGG

We performed GO enrichment analysis on related DEGs and conducted GO functional annotation analysis on differentially expressed genes, classifying them into three ontologies: biological processes, cellular components, and molecular functions.

To elucidate the physiological regulation of differentially expressed genes in the pathophysiological processes following infection, and to further understand the biological processes of these genes, we conducted Gene Ontology (GO) functional enrichment analysis. The top 30 GO enrichment functions of down-regulated and up-regulated genes are presented in Figure 4a–c. During the infection, up-regulated differentially expressed genes (DEGs) are involved in carbohydrate metabolic processes, extracellular regions, multicellular organism development, and structural constituents of carbohydrates, as well as ribosomal enrichment and serine-type endopeptidase activity. Conversely, down-regulated DEGs are primarily concentrated in extracellular regions and spaces, protein refolding, heme binding, iron ion binding, unfolded protein binding, and multicellular organism development. Infected *S. callosus* appear to exhibit increased metabolic activity, particularly concerning the regulation of cell growth, metabolism, and signaling activities.

Gene expression is a complex process involving the coordinated regulation of multiple genes to control specific functions in insects. Three treatment groups (T48 vs. cK, T96 vs. cK, and T144 vs. cK) exhibited enrichment in 270, 257, and 314 metabolic pathways, respectively. Significantly enriched pathways were identified based on a threshold of *p* < 0.05, with the top 20 pathways ranked in bubble figure.

KEGG enrichment analysis indicated that most differentially expressed genes (DEGs) were primarily concentrated in biological pathways closely associated with insect development and stress resistance (Figure 4d–f). Specifically, the most abundant pathways of DEGs included antigen processing and presentation, lysosomal function, endoplasmic reticulum protein processing, and ribosomal activity. In addition to these pathways, processes related to pancreatic secretion and cytochrome P450 enzymes were also noted. The estrogen signaling pathway, ABC transporter, apoptosis regulation, and glutathione metabolism were also notably concentrated. By separating the up-regulated and down-regulated genes and enriching them, we found that the up-regulated genes were primarily concentrated in transport and catabolism, carbohydrate metabolism, translation, digestive system, and signal transduction. It is noteworthy that the number of up-regulated genes enriched in the immune system generally increased with the progression of infection time. However, during the middle stages of infection, this number was partially suppressed, leading to a decline in the population. The estrogen signaling pathway, ABC transporter, apoptosis regulation, and glutathione metabolism were also notably concentrated. By separating the up-regulated and down-regulated genes and enriching them, we found that the up-regulated genes were primarily concentrated in transport and catabolism, carbohydrate metabolism, translation, digestive system, and signal transduction. These findings were consistent with the results of Gene Ontology (GO) enrichment analysis, indicating that infected *S. callosus* exhibited resistance to *B. bassiana* infection by enhancing metabolic activity and regulating more signal transduction pathways.

### 3.6. Metabolomic Analysis of the Response of S. callosus After Infection with B. bassiana

To further investigate the impact of *B. bassiana* on the response of physiological metabolizers in *S. callosus*, we employed UHPLC-MS/MS for metabolic profiling. Through PLS-DA analysis, a significant separation between ck and groups T48, T96, and T144 (Figure 5a) was observed, indicating substantial biochemical alterations in nodule tissue following infection. Moreover, 430, 390, and 451 metabolites were identified at different infection time points. Specifically, compared to the control group, group T48 exhibited 430 DEMs (229 upregulated and 201 downregulated), T96 showed 390 DEMs (200 upregulated and 190 downregulated), and T144 exhibited 451 DEMs (227 upregulated and 224 downregulated). Collectively, these findings suggest notable fluctuations in tissue metabolism as a response to *B. bassiana* infection.

The KEGG pathway enrichment analysis of differentially expressed metabolites (DEMs) from *S. callosus* revealed their involvement in diverse metabolic pathways to counteract *B. bassiana* infection at distinct time points. At 48 h post infection (group T48), the DEGs significantly perturbed central carbon metabolism in cancer, as well as glycine, serine, and threonine metabolism (*p* < 0.05). Additionally, alterations were observed in serine and threonine metabolism, d-amino acid metabolism, aminoacyl-tRNA biosynthesis, purine metabolism, and glycine and dicarboxylate metabolism, among other metabolic pathways (Figure 5b).

At 96 h post infection (group T96), ABC transporters, the two-component signaling system, and aminoacyl-tRNA biosynthesis were significantly altered (*p* < 0.05). Additionally, there were significant changes in the central carbon metabolism in cancer, choline metabolism in cancer, purine metabolism, d-amino acid metabolism, protein digestion and absorption, and other metabolic pathways (Figure 5c).

At 144 h post infection (group T144), *B. bassiana* significantly affected ABC transporters, aminoacyl-tRNA biosynthesis, pathways such as central carbon metabolism in cancer and choline metabolism in cancer (*p* < 0.05). Furthermore, there were notable alterations observed in purine metabolism; d-amino acid metabolism; glycine, serine, and threonine metabolism; and glycerophospholipid metabolism (Figure 5d).

### 3.7. Immune Response of S. callosus After Infection

The genes related to insect immunity have been reported previously [34,35,36], and the possible genes related to insect immunity were screened according to the annotated information from the NR database. A total of 331 immune-related genes were identified in the transcriptome data of *S. callosus*. The immune-related metabolites were identified based on the annotated information from the Human Metabolome Database (HMBD). At 48, 96, and 144 h post-infection, a total of 42, 50, and 43 immune-related metabolites were, respectively, screened. Relevant Appendix A contain comprehensive immune-related transcriptomic and metabolomic data.

Subsequently, these immune genes were subjected to enrichment analysis, revealing that at 48 h post infection, significant activation was observed in pathways associated with proteoglycans in cancer (ko05205), lysosome (ko04142), autophagy–animal pathway (ko04140), and fluid shear stress and atherosclerosis (ko05418). Furthermore, at 96 h post infection, there was significant activation of the Toll and Imd signaling pathway (ko04624), which is closely related to insect immunity. Although an ongoing immune response was detected at 144 h post infection within *S. callosus*, most of the pathways involved in this response had been inhibited. Notably, it can be inferred that the proteoglycans in cancer (ko05205) pathway remained consistently active following successful infection. Additionally, autophagy played a crucial role throughout the entire immune process (Figure 6a–f).

All immune-related metabolites screened can be broadly categorized into the following classes: glycerophospholipids, carboxylic acids and derivatives, organooxygen compounds, azolidines, prenol lipids, purine nucleosides, pyridines and derivatives, benzimidazole ribonucleosides and ribonucleotides, steroids and steroid derivatives, keto acids and derivatives, benzene and substituted derivatives, fatty acyls, imidazopyrimidines, glycinamide ribonucleotides, isoflavonoids, pyrimidine nucleosides, carboximidic acids and derivatives, flavonoids, triazines, quinolines and derivatives. KEGG enrichment analysis of these immune metabolites revealed that the glucagon signaling pathway (ko04922), cancer central carbon metabolism (ko05230), glycerophospholipid metabolism (ko00564), and cancer choline metabolism (ko05231) were identified as the most significantly enriched metabolic pathways associated with immune function. (Figure 6g–i). Upon infection, glycerophospholipids, carboxylic acids and derivatives, organooxygen compounds, keto acids and derivatives, and the immune metabolites of azolidines were significantly upregulated. For example, PC, l-aspartic acid, isocitric acid, n-acetyl-l-glutamic acid, d-glycerate 3-phosphate, 2-phospho-D-glyceric acid, D-arabinono-1,4-lactone, trehalose, 2-keto-6-acetamidocaproate, and hydantoin-5-propionic acid were among those significantly upregulated.

The immune metabolites in four metabolic pathways, i.e., glucagon signaling pathway, central carbon metabolism in cancer, glycerophospholipid metabolism, and choline metabolism in cancer, were traits, and the immune genes were analyzed WGCNA. Ultimately, the immune genes were categorized into four modules; however, the gray module did not exhibit any reference significance or belong to any specific module (Figure 7a). The Pearson correlation algorithm was employed to calculate the correlation coefficients and *p*-values between the characteristic genes and traits of the modules (Figure 7b). Modules related to each trait were identified using a threshold value of |r| > 0.8 for the correlation coefficient and a *p*-value < 0.05. Specifically, there was a significant positive correlation (r = 0.92, 0.92, 0.81; *p* < 0.05) between citric acid, isocitric acid, glycerophosphocholine and glycerylphosphorylethanolamine, and the blue module, while a significant negative correlation (r = −0.87, −0.91, −0.93; −0.92, *p* < 0.05) was observed with the turquoise module. Furthermore, PC(22:2(13Z,16Z)/0:0) exhibited a significant negative correlation with the blue module (r = −0.85; *p* < 0.05), whereas glycerophosphocholine showed a significant negative correlation with the turquoise module (r = −0.92; *p* < 0.05). Additionally, D-glycerate 3-phosphate and 2-phospho-d-glyceric acid demonstrated significantly positive correlations with the yellow module (r = 0.88, 0.92; *p* < 0.05).

The top 50 genes with the highest connectivity within each module were examined to reveal their relationships. The hub genes of each module were determined based on their strong connections with peripheral points. In the blue module, CYP4C1 (TRINITY_DN30634_c0_g1_i9_3) and CYP4g56 (TRINITY_DN23735_c0_g1_i3_3) emerged as hub genes (Figure 8a). Similarly, collectin-10-like (TRINITY_DN22704_c0_g2_i1_4) and CECR1/ADA2 (TRINITY_DN26550_c2_g2_i3_1) represented the hub genes in the turquoise module (Figure 8b). Lastly, CTSL (TRINITY_DN20512_c0_g1_i2_4 and TRINITY_DN27537_c1_g3_i1_2) were identified as hub genes within the yellow module (Figure 8c). Based on the aforementioned WGCNA analysis, it is evident that the central gene types include lectin C-type domain, cytochrome P450, and cysteine protease cathepsin L genes.

### 3.8. Integrative Analysis of Immune-Related Genes and Metabolites

The Pearson correlation coefficient between the top 20 differential genes and 20 differentially expressed metabolites was calculated, followed by the construction of a correlation cluster heat map. Notably, a strong correlation was observed between these variables (Figure 9a–c).

The changes in immune-related genes and metabolites of *S. callosus* during *B. bassiana* infection were analyzed and mapped to the KEGG pathway. The KEGG pathways shared by the two omics were plotted in a histogram (Figure 9d–f). The results indicate that the common pathways of related genes and metabolites mainly involve autophagy-animal, arachidonic acid metabolism, chemical carcinogenesis, metabolism of xenobiotics by cytochrome P450, glycerophospholipid metabolism, pathogenic Escherichia coli infection, vitamin digestion and absorption, longevity regulating pathway-worm, pentose and glucuronate interconversions, fatty acid biosynthesis, arginine biosynthesis, thyroid hormone synthesis, vitamin B6 metabolism, glycine, serine, and threonine metabolism, and glutathione metabolism. The enrichment of differentially expressed metabolites (DEMs) and differentially expressed genes (DEGs) both highlight autophagy–animal and arachidonic acid metabolism, which occur at high frequencies across different infection periods, prompting further investigation into these two common pathways. Additional pathway details are listed in Appendix A. A simplified version of the signaling pathway networks for autophagy-animal (Figure 10) and arachidonic acid metabolism (Figure 11) in KEGG was plotted. We found that the PTEN, TSC2, PKA, TP53INP2, LC3, and LAMP genes play crucial roles in the autophagy-animal pathway, jointly regulating the production of cathepsin and degradation of the innerveside. In arachidonic acid metabolism, PLA2G, HPGDS, PTGES, and CBR co-regulate the final production of 15-keto-PGF2α and 11-epi-PGF2α. The involvement of PLA2G, specifically, assumes a pivotal role in orchestrating this intricate process. The synthesis of downstream PGs is mediated by PLA2G.

### 3.9. RT-qPCR Validation of Transcriptome Sequencing Data

To authenticate the transcriptome sequencing results, we randomly selected six immune genes with differential expression (Figure 12). The trends observed in the transcriptome sequencing data were consistent with those validated by RT-qPCR, affirming the higher reliability of our transcriptome sequencing data.

## 4. Discussion

In the present study, transcriptome and metabolome analyses were employed to investigate alterations in the gene expression and metabolism of *S. callosus* following infection with *B. bassiana*. A total of 9933 upregulated genes and 6687 downregulated genes were identified, with the number of upregulated genes exhibiting significant temporal variation during infection, indicating substantial changes in gene expression and metabolites induced by *B. bassiana* in *S. callosus*. The Gene Ontology (GO) enrichment analysis revealed prominent pathways associated with digestion, oxidase activity, and collagen decomposition, among others. Additionally, the KEGG enrichment analysis demonstrated that differentially expressed genes were primarily enriched in cytochrome P450, lysosome function, apoptosis regulation, phagosome formation, glutathione metabolism, and cysteine and methionine metabolism. Metabolites were predominantly enriched in central carbon metabolism, glycine, serine and threonine metabolism, purine metabolism, and the ABC transport system in cancer.

According to the WGCNA analysis, the key gene types involved in immune response were lectin C-type domains, cytochrome P450 enzymes, and cysteine proteinase cathepsin L. Cathepsins are widely distributed proteolytic enzymes found in various biological tissues [37]. The involvement of cathepsin L in numerous physiological processes in vivo, including the regulation of the immune response and tumor dissemination, has been well documented [38,39,40]. As a multifunctional enzyme, insect cathepsin is also involved in the insect response to external stimuli [41]. Insects exhibit a downregulation of the expression of certain cathepsin genes in response to sublethal concentrations of pesticide-induced stress [42]. The expression of cathepsin was significantly upregulated in citrus psyllid under high temperature stress [43]. The expression of the cathepsin gene of *Monochamus alternatus* was downregulated after treatment with a sublethal concentration of *Bacillus thuringiensis* [44]. In this study, the expression of genes of cathepsin L varied, potentially due to its distinct biological functions in cellular immunity and humoral immunity. In the innate immune response of insects, the precise recognition of invasive pathogens is crucial [45]. The immune response relies on important recognition receptors that control the recognition process, with C-type lectin being a common pattern recognition receptor [46]. C-type lectins serve as upstream pattern recognition receptors capable of recognizing various microorganisms, and also function as downstream effector factors with bactericidal activity, even sensing cancer cell death [47,48]. In this study, the upregulation of the C-type lectin gene in the blue and turquoise modules was observed at 48 h post-infection, indicating rapid pathogenic microorganism recognition and subsequent immune response activation by *S. callosus* immune system when invaded by *B. bassiana*. The association between the high expression of the P450 gene and elevated levels of metabolic resistance to various pesticides in insects has been extensively demonstrated by numerous studies [49,50]. In this study, significantly higher expression levels of the cytochrome P450 gene were observed in the yellow and turquoise modules compared to the ck group, suggesting its potential involvement in metabolic detoxification. The humoral immunity in insects encompasses the classical Toll pathway, Imd pathway, and JSK/STAT pathway. Upon fungal invasion and entry into the hemolymph, insect pattern recognition receptors identify antigenic substances on the surface of invading pathogens and initiate gene activation associated with immune pathways. The Toll pathway directly facilitates the expression of AMP genes and enhances the survival and proliferation of immune cells in certain insect species [51]. However, transcriptome analyses have revealed that this pathway not only promotes AMP expression, but also releases small antifungal molecules or triggers the production of other substances crucial for immune response [52,53]. The Toll and IMD pathways were significantly activated at 96 h after *B. bassiana* infection, at which time humoral immunity had begun to play a role.

Autophagy is a catalytic process activated in response to inadequate nutrition and plays a crucial role in various physiological and pathological processes, including stress, neurodegenerative diseases, infections, and cancer. The mTOR kinase acts as a pivotal regulatory molecule in the induction of autophagy by positively promoting its activation while negatively inhibiting it. Serine/threonine kinases can function downstream of the mTOR complex [54,55,56]. Following 48 h of infection, there was a significant activation of the autophagy pathway in *S. callosus*. The infection impacted the digestive system of *S. callosus*, leading to limited nutrient acquisition and the subsequent activation of autophagy. This process could be considered an adaptive mechanism for survival in *S. callosus*. The detection of serine/threonine-related metabolites at this time further supports the congruence between the metabolic process and the transcriptome results. Body immunity and metabolic homeostasis are intricately linked, with numerous diseases capable of inducing metabolic disorders. Our study reveals an enrichment of immune metabolites in the glucagon pathway, predominantly characterized by downregulated metabolites. The glucagon pathway operates downstream of the insulin signaling pathway, which plays a central role in metabolic regulation during infection and inflammation [57]. The activation of the islet receptor induces the AKT pathway to exert negative control over fat and glycogen storage [58].

Eicosanoic acid plays a crucial role in the immune response of insects. In animals, phospholipase A2 (PLA2) catalyzes the release of arachidonic acid (AA) from phospholipids (PLs) for the biosynthesis of various eicosanoids [59]. Prostaglandins (PGs) serve as fundamental mediators of cellular and humoral immunity in insects, stimulating the formation of blood nodules and cysts in response to microbial infection [60]. Prostaglandins (PG), particularly PGE2, can induce specific immune responses, such as the regulation of blood cell diffusion [61], and can also promote the expression of the antimicrobial peptide AMP [62]. Additionally, PG mediates the release of inactive prophenoloxidase into the plasma, where it is subsequently activated by serine proteases through G protein-coupled receptors on the cell surface [61,63]. In this study, it is observed that phosphatidylcholine (PC) is converted into arachidonic acid by phospholipase A2 group G (PLA2G). When PLA2G is upregulated, downstream enzymes HPGDS, PTGES, and CBR are also upregulated, resulting in the formation of PGF. After binding to specific receptors, prostaglandins regulate various cellular activities including cell proliferation, differentiation, apoptosis, platelet aggregation, and other crucial processes.

## 5. Conclusions

In this study, the transcriptomics and metabolomics of *S. callosus*. following infection by *B. bassiana* were investigated. The results of GO and KEGG analyses indicated that infection by *B. bassiana* enhanced signal transduction between cells and accelerated metabolism in *S. callosus*.. Concurrently, detoxification enzymes were synthesized to metabolize the toxins secreted by *B. bassiana*, thereby resisting further infection. Metabolomics analysis confirmed that glycerophospholipids, carboxylic acids and derivatives, organic oxygen compounds, ketoacids and derivatives, as well as azane immune metabolites, were significantly upregulated following infection by *B. bassiana*. GSEA analysis revealed that during the early stages of infection, pathways such as proteoglycans in cancer (ko05205), lysosome (ko04142), autophagy–animal pathway (ko04140), and fluid shear stress and atherosclerosis (ko05418) were significantly activated. The toll and imd signaling pathways (ko04624) were activated during the mid-stage of infection (96 h post-infection), while a limited number of immune responses persisted during the later stages of infection. Transcriptional and metabolomic analyses indicated that autophagy and arachidonic acid metabolism played a crucial role in the overall immune process of *S. callosus*.

## Figures and Tables

**Figure 1 insects-15-00940-f001:**
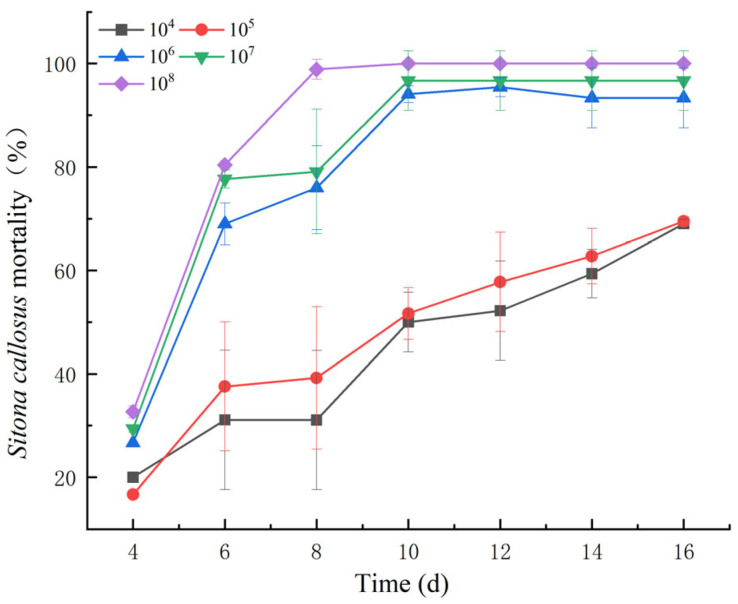
Impact of varying spore concentration treatments on the mortality of adult *S. callosus*.

**Figure 2 insects-15-00940-f002:**
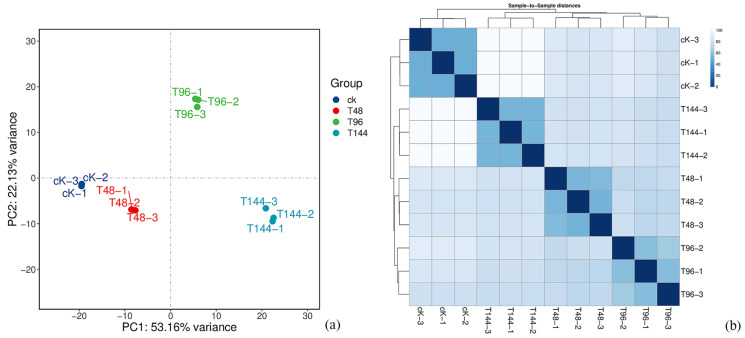
The repeatability of the transcriptome data from *S. callosus* was assessed through PCA and cluster analysis on the samples.

**Figure 3 insects-15-00940-f003:**
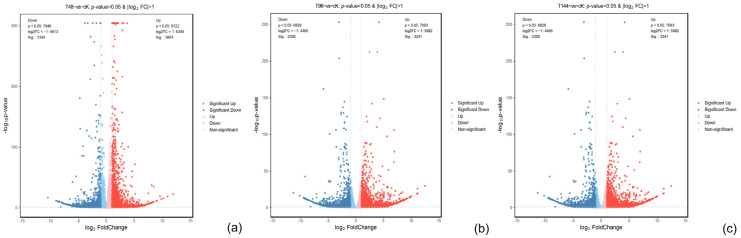
Differential expression gene volcano for *S. callosus* samples. Volcano plot illustrating differential gene expression in samples (**a**–**c**) at 48 h, 96 h, and 144 h post infection with *B*. *bassiana.* Blue dots represent downregulated genes while red dots indicate upregulated genes.

**Figure 4 insects-15-00940-f004:**
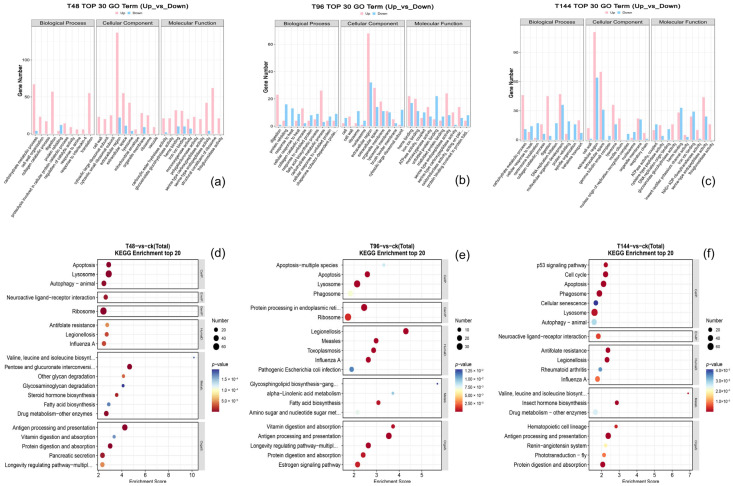
GO and KEGG enrichment analysis of differentially expressed genes. (**a**,**d**), Samples were infected with *B. bassiana* for 48 h (T48 vs. cK). (**b**,**e**), Samples were infected with *B. bassiana* for 96 h (T96 vs. cK). (**c**,**f**), Samples were infected with *B. bassiana* for 144 h (T144 vs. cK).

**Figure 5 insects-15-00940-f005:**
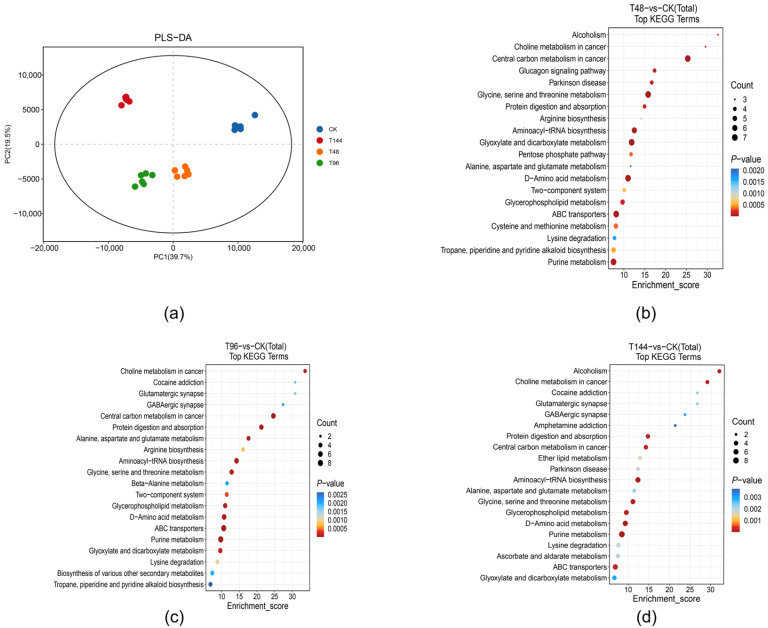
(**a**) Differential metabolite PCA and KEGG enrichment analysis. (**b**) Samples were infected with *B. bassiana* for 48 h (T48 vs. ck). (**c**) Samples were infected with *B. bassiana* for 96 h (T96 vs. ck). (**d**) Samples were infected with *B. bassiana* for 144 h (T144 vs. ck).

**Figure 6 insects-15-00940-f006:**
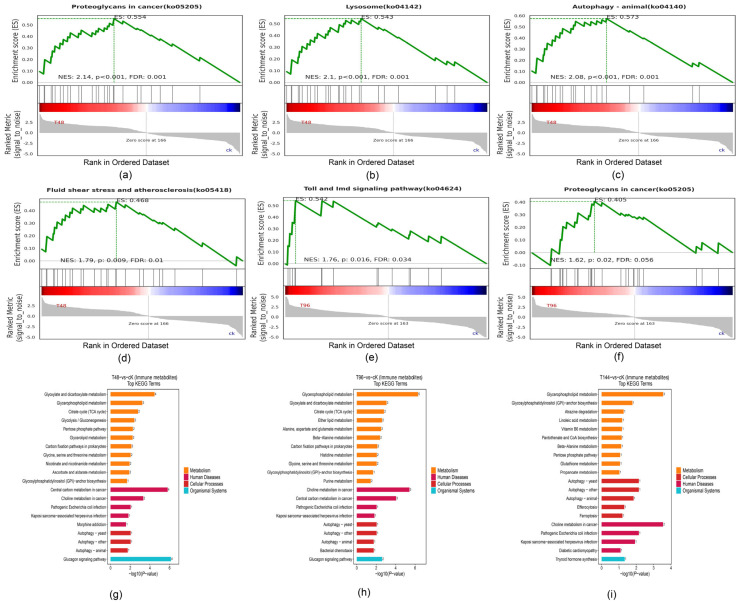
Enrichment analysis of immune gene GESA and immune metabolite KEGG. In GESA enrichment analysis, NES > 1, FDR < 0.25, and *p* < 0.05 were considered significant activation of the pathway. (**a**–**d**) After 48 h of infection, the proteoglycans in cancer (ko05205), lysosome (ko04142), Autophagy-Animal pathway (ko04140), and fluid shear stress and atherosclerosis pathway (ko05418) were significantly activated. (**e**) At 96 h after infection, the Toll and Imd signaling pathway (ko04624) exhibited significant activation. (**f**) The proteoglycans in cancer (ko05205) showed significant activation at 144 h after infection. (**g**) Enrichment analysis of immune metabolites based on the KEGG database was conducted after 48 h of infection with *B. bassiana.* (T48 vs. ck). (**h**) Enrichment analysis of immune metabolites based on the KEGG database was conducted after 48 h of infection with *B. bassiana*. (T96 vs. ck). (**i**) Enrichment analysis of immune metabolites based on the KEGG database was conducted after 48 h of infection with *B. bassiana*. (T144 vs. ck).

**Figure 7 insects-15-00940-f007:**
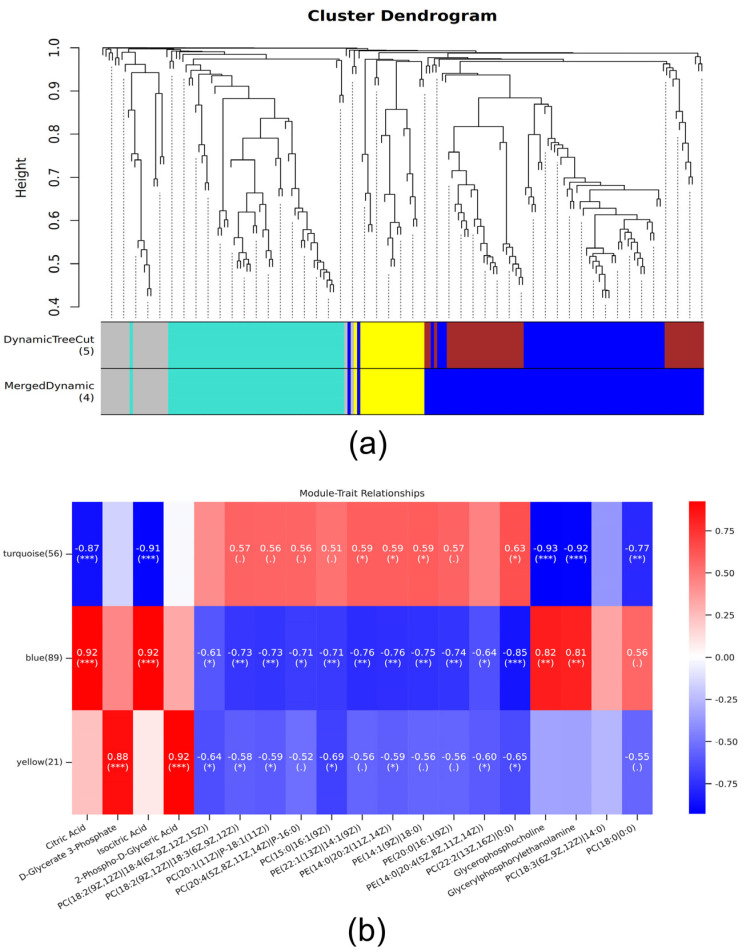
Module identification and trait correlation analysis in WGCNA analysis. (**a**) The immune genes were categorized into four modules; the gray module did not exhibit any reference significance or belong to any specific module. (**b**) is correlation analysis between traits and each module. Blue represents negative correlation, red represents positive correlation. The statistical significance levels for the metabolites and modules in the figure are denoted as *, **, and ***, corresponding to *p* < 0.05, *p* < 0.01, and *p* < 0.001, respectively.

**Figure 8 insects-15-00940-f008:**
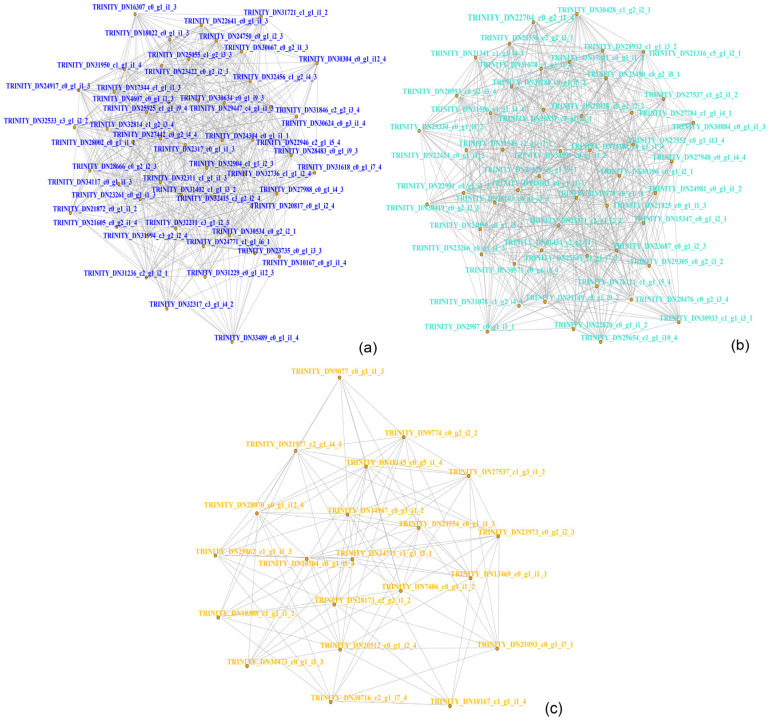
WGCNA-hub gene analysis of immune genes based on immune metabolites. (**a**–**c**) represent the hub genes of the blue, turquoise, and yellow modules, respectively. The more connections a gene has to its periphery, the more central it is.

**Figure 9 insects-15-00940-f009:**
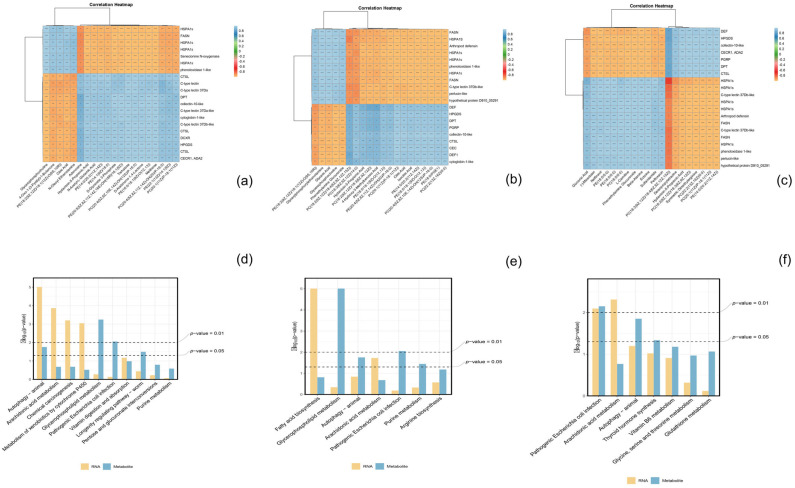
Correlation-based cluster heat maps depicting immune-related DEGs and DEMs, along with their co-enriched KEGG pathways. In (**a**–**c**), the color orange exhibits a negative correlation, while the color blue demonstrates a positive correlation. In (**d**–**f**), the yellow bars denote ribonucleic acid (RNA), while the blue bars denote metabolites. (**a**,**d**) Samples were infected with *B. bassiana* for 48 h (T48 vs. cK). (**b**,**e**) Samples were infected with *B. bassiana* for 96 h (T96 vs. cK). (**c**,**f**) Samples were infected with *B. bassiana* for 144 h (T144 vs. cK). The statistical significance level of the correlation between metabolites and genes in the figure was denoted by *, **, and ***, corresponding to *p* < 0.05, *p* < 0.01, and *p* < 0.001 respectively.

**Figure 10 insects-15-00940-f010:**
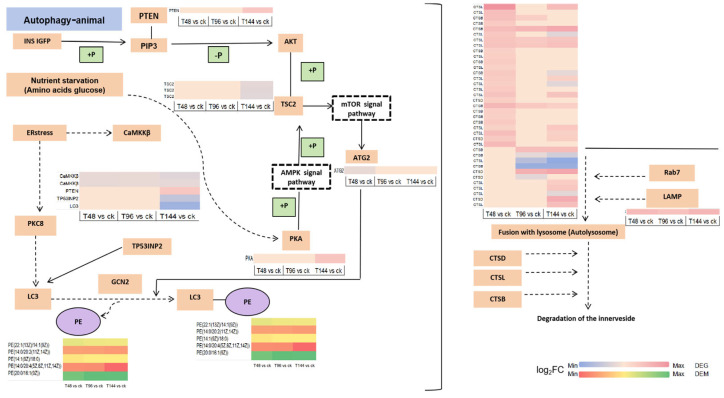
Autophagy–animal pathway. The purple ovals represent metabolites, and the yellow rectangles represent genes annotated in this pathway. The expression profiles adjacent to the genes and metabolites are heat maps generated using log2 fold change (log_2_FC) values. The original KEGG pathway can be found in the Appendix A.

**Figure 11 insects-15-00940-f011:**
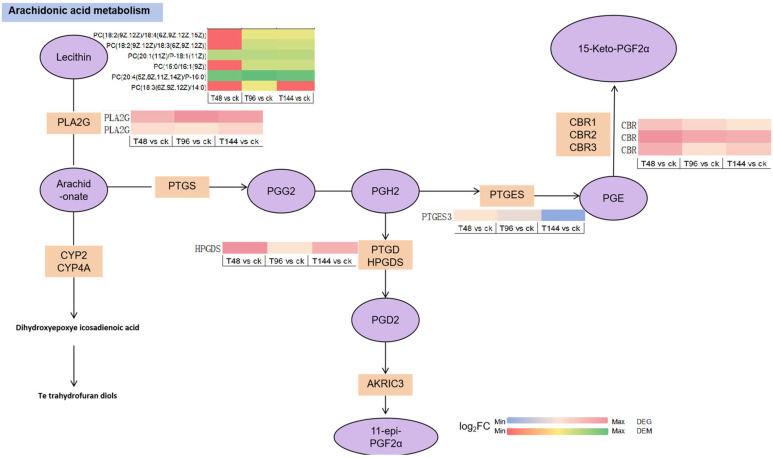
Arachidonic acid metabolism pathway. The purple ovals represent metabolites, while the yellow rectangles represent genes annotated in this pathway. The expression profiles adjacent to the genes and metabolites are heat maps generated using log_2_ fold change (log_2_FC) values. The original KEGG pathway can be found in the Appendix A.

**Figure 12 insects-15-00940-f012:**
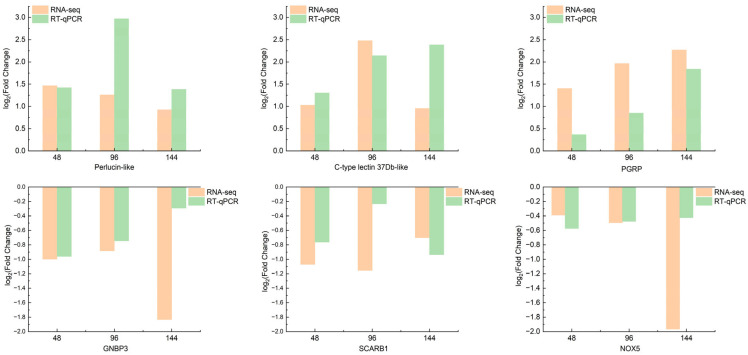
Comparison of differentially expressed gene transcriptome and RT-qPCR results.

## Data Availability

The transcriptomic raw data related to this thesis have been uploaded to NCBI (https://www.ncbi.nlm.nih.gov/geo/query/acc.cgi?acc=GSE281105) under the accession number GSE281105 (accessed on 10 November 2024). The metabolomic raw data have been uploaded to the China National Center for Bioinformation (CNCB) (https://ngdc.cncb.ac.cn/gsub/submit/bioproject/subPRO047171/overview) under submission number subPRO047171(accessed on 13 November 2024).

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
