# Peer review of "Revealing the Immune Response of Sitona callosus Gyllenhal to Entomopathogenic Fungi Beauveria bassiana Infection Through Integrative Analyses of Transcriptomics and Metabolomics"

_insects, 2024, doi:10.3390/insects15120940_

Round 1
Reviewer 1 Report
Comments and Suggestions for Authors
I genuinely enjoyed reading the submitted manuscript “Revealing the Immune Response of Sitona callosus Gy11. to Beauveria bassiana Infection through Integrative Analyses of Transcriptomics and Metabolomics”. The study is scientifically sound and undoubtedly presents and important and interesting story. This being said, I feel that the full potential of the data is not presented in the current form of the manuscript. The data analysis procedures need a bit of clarification, the conclusions might have been more precise, and figures have room for improvement. Please find more particular comments below.
== Principal comments / concerns
1. Was the object of the study Sitona cylindricolli or S. callosus? I am not an expert in Coleoptera, but it does not seem a good idea to use two different species names interchangeably. What is the differenece between the two and which is correct?
2. I believe the correlation or intersection between transcriptomic and metabolomic responses could be made better. So far, I can only see the correlation in expression values (Fig. 8), the scientific value of which is at unclear to me (see below) and Figs. 10-12 devoted to particular pre-chosen groups of proteins. I would suggest to add an analysis of based intersection of some overrepresented function annotation terms, most obviously KEGG pathways, between transcriptomic and metabolomic data. It’s clear that both data sets are quite noisy inherently, but it’s possible that such a restrictive comparison might be very helpful in revealing the biological information.
3. As mentioned already, the study is scientifically sound and the study object undoubtedly important, but the conclusions are unfortunately quite weak. What reads from them is that there is immune response in the studied pest to an entomopathogenic fungus. This is absolutely expected and could have been guessed without the actual study. Please understand me correctly: I do not try to say in any way the study was unnecessary or is pointless. Instead, I would like to motivate the authors to formulate what was unusual, novel and unexpected about their results and how the results could be used to actually improve pest control. This would dramatically improve the impact of the manuscript for the field.
4. I could not find any indication about the raw data generated in this study being submitted to relevant databases (NCBI in the case of transcriptome sequencing and at least one of metabolome-related database for metabolites). It is the right of the authors to keep the information private until the publication of the manuscript, but please submit them and provide a reviewer’s token in this case.
As the most time effective way to publish the data, I would recommend the NCBI GEO database. It allows for the deposition of both raw and processed data (as well as assemblies), processes the data quite rapidly and allows for the generation of a reviewer’s token. On top of that, it would be ideal to also submit the assembly to the TSA / GenBank database, so that it is subsequently included into the databases used by BLAST toolkit.
== More particular textual comments
5. Title: would it make sense to add descriptions like “alfalfa pest” and “insect-controlling fungus” (or any other formulation) to the title to make it more comprehensive for general reader?
6. Title: what is “Gy11”? It seems like Gyllenhall, who is the author of the species, but it should be ll instead of 11 then. Moreover, please see the comment about the species name above.
7. Abstract: which immune-related metabolites?
8. L112-114 “The Beauveria bassiana strain B1 used in this study was isolated from the cadaver of Sitona cylindricollis, a pest of alfalfa, and showed low virulence against the pest in prelim-
inary tests.”: is this information published? If yes, pleas provide reference to publication. If no, please add more specific details in the results section.
9. L117 “Beauveria bassiana was cultured on PDA medium for approximately 10 days.”: medium recipe and/or reference? While potato dextrose agar is a standard media for fungal growth, there are multiple way to prepare it.
10. L134 “Total RNA was extracted from S. cylindricollis by total RNA isolation kit.”: which kit was used?
11. L139 “based on the similarity”: similarity to what?
12. L149-150 “R (v 3.2.0) was used to draw the column diagram and bubble diagram of the significant enrichment term.”: please add the relevant packages used and reference the publications if present. The authors of the packages deserve recoginition and citation.
13. L154-155 “The metabolite analysis of S. cylindricollis was completed by Shanghai Luming Biotechnology Co., Ltd. (Shanghai).”: i understand the analysis was outsourced, but the details are still needed... What were the particular material for extraction, extraction medium and procedure?
14. L166-170 “S. cylindricollis cDNA was extracted at each stage of infection, and the relative expression was determined using the 2−∆∆Ct method.”: first, I believe total RNA was extracted with subsequent cDNA synthesis, and second, please add a reference to the method.
15. L173-178 “survival rate”: if you’re speaking about survival rate (not mortality), maybe show the other way round (100-mortality, i.e. survival)?
In addition, I couldn’t find any information on the data analysis procedure in the methods section. It looks like Cox regression was applied, but I can’t be sure.
15. L233 “donkey bean” and L214 “4 groups of donkey root nodules”: what are those terms? I am not an expert in in the object, but couldn’t easily find it on the Internet either. If it’s a specific local term, please consider explaining it clearly or substituting with a broader used equivalent.
16. L239-240 “In the ck-vs-
T48, ck-vs-T96, and ck-vs-T144 groups, a total of 4188, 3387, and 9045 DEGs.”: this information should have been presented earlier (in the previous section).
17. L243-245 “Cel-
lular components mainly concentrated on organelle components, cells and cell compo-
nents, as well as protein complexes.”
AND
L251-252 “Cell components mainly include cells, cell components, organelle components etc.”: this seems is very unspecific and might actually be omitted
18. The transcript names on Figure 3 do not seem to add much information, as there is nothing other than the names (like function or sequences). Depending on what is the main piece of information the authors would like to convey with this figure, it might be a good idea to, for example, highligh the transcripts common for all three time points or omit the transcript IDs altogether.
18. Figure 4 does not seem to be highly informative. The only thing that is obvious from this illustration that there are many overrepresented terms. I do not have a clear solution for this problem, it might also be helpful to try to highlight either common transcripts or immunity-related transcripts. The same suggestion goes for Fig. 5 (even though it is a bit easier to decipher).
19. Figure 7a-f: what does the horizontal scale mean?
20. L356 and the whole section 3.7: I do not quite catch the concept of reenrichment analysis. If first immunity-related genes were chosen, what happened to the gene lists afterwards?
21. L360-361: Fig. 8 “The Pearson correlation coefficient between the top 20 differential genes and 20 differentially expressed metabolites”: correlation of what? If the analysis correlated the expression levels, then yes, the correlation is expected due to is purely mathematical reasons, as you correlate the top changed values.
Or correlated transcript / metabolite pair, e.g. glycerolphosphocholine and Trinity_DN32415_c3_g2_i2_4, have something in common? Then maybe use functional annotation of transcripts in addition to / instead of their names.
Please explain the idea of this analysis.
22. Fig. 9 is unfortunately unreadable. So far the only clear thing is there are some clusters. The version of the figure in the raw figure files is much better, but please consider increasing font sizes. There is a lot of white space to do so.
23. L384-386 “It is evident that the hub gene types are lectin C-type domain, cytochrome P450, heat shock genes and cysteine proteinase cathepsin L genes.”: some aspects are unclear. What is it the evidence source? Which transcript corresponds to which function? Where can the reader get the sequences for futher use?
24. Figure 10
There around 30 transcripts in the left and two marked at the righ. I cannot understand the connection between left and right parts. The same applies to Figs. 11 and 12. Would it be possible to reflect is somehow clearer?
In addition, is it possible to provide direct links to the pathway maps and/or spell out the abbreviations? Most of the readers wouldn’t probably remember most of the abbreviations by heart.
25. Fig. 13: where is transcriptome and where is qPCR? This all looks like qPCR, is it possible to add transcriptome data? For example, as different shading of the bars?
Comments on the Quality of English Language
There are a quite some few typos or unclear sentences throughout the text. In most cases they do not interfere with reading, but a proofreading round could really uplevel the text. Please find some examples below, but bear in mind that the list is by no means exhaustive.
L26 “differential expressed immune-related genes”: “differentially expressed immune-related genes”
L28-29 “Based on immune metabolites data using WGCNA method for analysis of immune-re-
lated genes allowed us to identify core genes involved at different stages of immunity induction.”: the two parts of the sentence do not agree
Something’s going on with the square brackets for references:
L46 “plant growth and development[7-11]” and multiple other places: space missing
L58-59 “programs.[13]”: space missing and period should be after the closing bracket.
L62-63 “infrequent[[16]”: space missing and excessive bracket.
L73 “by the baculovirus Autographa californica multiple nucleopolyhedro virus” looks overcomplicated.
L134-135 “Illumina Novaseq 6000 sequencing platform.”: broken sentence
L139 “was choose as”: “was chosen as”
L183 “High-throughput was employed”: a noun is missing
L188 “were aligned to Unigene”: “were aligned to assembled unigenes”? / “L188 “were aligned to Unigene”: “were aligned to unigenes foun in the assembly”?
L189-190 “Using Trinity[26] software for assembly, the results are shown in Table S2.,”: “The characteristics of assembly with Trinity are shown in Table S2.”?
L136 “adaptor”: “adapter”?
L193 “These results the accuracy of the transcriptome data ”: a verb is missing
L204 “, further analysis”: a verb is missing (allowing for?)
L209 “were annotated to the NR database,”: to?
L217-218 “When q-value<0.05, |log2FC|>1.0, we considered that this gene was significantly dif-
ferent among the groups, so as to select less differentially expressed genes.”: ... reformulate
L317 “B. bassiana” italics missing
L372-373 “The immune metabolites in four metabolic
pathways of cancer were traits, and the immune genes were analyzed by weighted gene
co-expression network analysis.”: I cannot see correlation between the first and second parts of the sentence?
L392 “pearson”: “Pearson”
L445 “cathepsin L.”: “cathepsin L”
L453 “In this study, In this study,”
L504 “The four immune core
genes selected were Lectin C-type domain, Cytochrome P450, heat shock gene, and Cys-
teine proteinase cathepsin L.”:
- first, capital letters unnecessary
- second, selected for what?
Reviewer 2 Report
Comments and Suggestions for Authors
Review Report
This study investigates the immune response mechanisms in Sitona callosus, an insect pest of alfalfa, following infection with the entomopathogenic fungus Beauveria bassiana. Using transcriptomic and metabolomic analyses, the authors examine changes in gene expression and metabolic processes over three infection time points: 48, 96, and 144 hours. The study identifies various immune response pathways, including carbohydrate metabolism, cytochrome P450 activity, lysosome function, apoptosis regulation, and pathogen response pathways. By correlating transcriptomic and metabolomic data, the study highlights key immune-related genes and metabolites, offering valuable insights into potential biological control strategies for managing S. callosus.
The research is timely and addresses an important issue in agricultural pest management. The use of both transcriptomics and metabolomics allows for a more comprehensive understanding of the host immune response, and the study’s findings hold potential for enhancing biocontrol strategies targeting S. callosus. However, there are some areas where the manuscript could be improved for greater clarity, depth, and rigor in interpretation.
Comments for authors
The Materials and Methods section of this manuscript has several areas for improvement to enhance clarity, reproducibility, and overall rigor. Here are some identified shortcomings:
1. The Beauveria bassiana strain description lacks information on its maintenance conditions, such as specific temperature, humidity, or any conditions that might affect its virulence.
2. The RNA extraction process would benefit from more detail, such as specifying RNA extraction kit brand and exact protocols to improve reproducibility.
3. The sequencing platform and parameters are mentioned, but there is no information on the depth of sequencing (e.g., number of reads, coverage), which is critical for assessing the robustness of transcriptomic data.
4. The metabolomics analysis references “widely untargeted” methods without specifying the type of instrument or the exact metabolite extraction protocols, which are necessary to understand the range and sensitivity of metabolite detection.
5. Criteria used for selecting metabolites of interest, such as “VIP≥1,” should be explained briefly for readers unfamiliar with this threshold.
6. Specific statistical tests are mentioned (e.g., PCA, PLS-DA), but the authors do not describe which parameters they used for model validation or if any cross-validation techniques were employed to confirm the reliability of multivariate analysis results.
7. In section 2.5, where qRT-PCR is mentioned, further details on how the data were statistically analyzed (e.g., software used, significance thresholds) are lacking, and it’s unclear how many replicates were used for each gene validation.
Enhancing these areas would improve the study's transparency and reproducibility, making it easier for other researchers to replicate and build upon this work. I recommend a major revision.
Round 2
Reviewer 1 Report
Comments and Suggestions for Authors
The authors did a great job addressing my questions and concerns! I'd like to commend them for taking critique seriously and wish the best of luck in continuing this work.
One small comment (still on the general question of citing the tools used):
L166-167 "GO enrichment and KEGG pathway enrichment analysis of the DEGs were performed, respectively, using R based on the hypergeometric distribution": please check if only the base R package was used or another specialized package was employed. To me, it looks like fgsea or clusterprofiler. If additional packages were indeed used, please consider citing them.
Author Response
L166-167 "GO enrichment and KEGG pathway enrichment analysis of the DEGs were performed, respectively, using R based on the hypergeometric distribution": please check if only the base R package was used or another specialized package was employed. To me, it looks like fgsea or clusterprofiler. If additional packages were indeed used, please consider citing them.
Response: Thank you for recognizing our work, and I appreciate your meticulous review again. Yes, the 'clusterProfiler' package is utilized in R for the analysis. I have included the relevant literature citations in the manuscript.

Reviewer 2 Report
Comments and Suggestions for Authors
The authors of the article have addressed all my questions. I have no further comments and therefore recommend that the article be accepted for publication in the journal.
Author Response
Thank you once again for your meticulous review and for acknowledging our work. I wish you all the best.